# Exosomes in Alpha-Synucleinopathies: Propagators of Pathology or Potential Candidates for Nanotherapeutics?

**DOI:** 10.3390/biom12070957

**Published:** 2022-07-08

**Authors:** Panagiota Mavroeidi, Maria Vetsi, Dimitra Dionysopoulou, Maria Xilouri

**Affiliations:** Center of Clinical Research, Experimental Surgery and Translational Research, Biomedical Research Foundation of the Academy of Athens, 11527 Athens, Greece; pmavroeidi@bioacademy.gr (P.M.); mariavetsi612@gmail.com (M.V.); dimidion@gmail.com (D.D.)

**Keywords:** alpha-synuclein, biomarkers, exosomes, glia, neurons

## Abstract

The pathological accumulation of alpha-synuclein governs the pathogenesis of neurodegenerative disorders, such as Parkinson’s disease, dementia with Lewy bodies, and multiple system atrophy, collectively termed alpha-synucleinopathies. Alpha-synuclein can be released in the extracellular space, partly via exosomes, and this extracellular protein pool may contribute to disease progression by facilitating the spread of pathological alpha-synuclein or activating immune cells. The content of exosomes depends on their origin and includes specific proteins, lipids, functional mRNAs and various non-coding RNAs. Given their ability to mediate intercellular communication via the transport of multilevel information, exosomes are considered to be transporters of toxic agents. Beyond neurons, glial cells also release exosomes, which may contain inflammatory molecules and this glia-to-neuron or neuron-to-glia transmission of exosomal alpha-synuclein may contribute to the propagation of pathology and neuroinflammation throughout the brain. In addition, as their content varies as per their originating and recipient cells, these vesicles can be utilized as a diagnostic biomarker for early disease detection, whereas targeted exosomes may be used as scaffolds to deliver therapeutic agents into the brain. This review summarizes the current knowledge regarding the role of exosomes in the progression of alpha-synuclein-related pathology and their potential use as biomarkers and nanotherapeutics in alpha-synucleinopathies.

## 1. Introduction

Alpha-synucleinopathies are a class of neurodegenerative disorders that are neuropathologically characterized by the pathological deposition of the intrinsically disordered pre-synaptic neuronal protein alpha-synuclein (αSyn). The protein accumulates in neurons in Parkinson’s disease (PD) and dementia with Lewy bodies (DLB) forming the Lewy bodies (LBs) and Lewy neurites (LNs), whereas in multiple system atrophy (MSA) αSyn aggregates are formed mainly within oligodendrocytes, forming the glial cytoplasmic inclusions (GCIs). Multiple toxic conformations of the protein, varying from soluble monomers to insoluble fibrillar forms, exist in the brain and are able to propagate in a prion-like manner from cell to cell, resulting in the pathological progression of the disease [1,2]. Thus, research in the field is focusing on uncovering the mechanisms that underlie αSyn aggregation and transmission as it represents a crucial step towards early diagnosis and effective disease treatment. Until recently, the main concept related to αSyn toxicity was that the misfolded forms of the protein leading to neurodegeneration are limited intracellularly. However, this concept was questioned when accumulating evidence showed that αSyn can be found extracellularly in human plasma and cerebrospinal fluid (CSF) of patients with PD [3,4]. Subsequent studies supported that both monomeric and aggregated αSyn can be secreted from neuronal cells either via vesicles or exosomes and that exosome-associated αSyn can exert various deleterious effects on neighboring cells [5,6,7]. Exosomes have been associated with prion transmission from infected neuronal donor cell lines to healthy recipient cells [8,9] thus placing the study of exosomes at the forefront of the neurodegenerative diseases field.

On the other hand, it has been demonstrated that exosomes could provide neuroprotection via the externalization of the increased αSyn load that counterbalances the elevated intracellular αSyn levels [10,11]. Interestingly, exosomes derived from glial cells could transport to and be taken-up by neurons, which may either be beneficial or detrimental to neurodegenerative diseases. In particular, activated glial-derived exosomes may not only spread αSyn pathology but also deliver and transmit pro-inflammatory mediators from glia-to-glia or glia-to-neurons, leading to the propagation of the inflammatory response and contributing thus to neuronal degeneration and disease progression. Microglial-derived exosomes have been reported to exert mainly neurotoxic effects as they can facilitate αSyn transmission in the brain [12,13], in contrast to the astroglial-derived exosomes that have been reported to exert neuroprotective effects [14]. As for oligodendroglial-derived exosomes, it has been recently suggested that their reduced secretion may be related to pathological αSyn aggregation in MSA [15]. Given that exosomal content depends on the origin cell, the exosomal protein cargo and its modifications upon disease pathology render the study of exosomes as potential biomarkers, a novel and rapidly evolving field within neurodegenerative disease diagnostics. In the sections below, we discuss the current knowledge regarding the role of exosomes in the neuron-glia communication both under physiological and pathological conditions and provide existing evidence related to the potential use of exosome-based approaches as candidate therapeutics for alpha-synucleinopathies.

## 2. Formation of Exosomes and Physiological Role

### 2.1. Exosome Biogenesis

Exosomes are small extracellular vesicles with a diameter of 40–100 nm originating from endosomal compartments called multivesicular bodies (MVBs) consisting of various intraluminal vesicles (ILVs). Exosomes are known to be present in plasma, serum, urine, CSF, saliva, breast milk and other secretions [16] and are able to shuttle protein and genetic material cargo between neighboring and distant cells [17,18]. Most of the cell types found in the brain, such as neurons, astrocytes, oligodendrocytes and microglia, have all been shown to secrete exosomes both in vitro and in vivo [19,20,21]. Exosomal cargo, which relates to the origin cell, comprises of lipids, proteins, mRNA, small regulatory RNAs and signaling molecules implicated in the regulation of gene expression and protein activity of recipient cells [22]. The main protein markers present in all exosomes deriving from different cell types are Alix, the tetraspanins CD63 and CD81 present on the vesicle surface and the heat shock HSP-70 and HSP-90 proteins [22]. Other cell-specific proteins characterize the origin of exosomes, such as the major myelin proteolipid protein (PLP), which is enriched in oligodendrocyte-derived exosomes [23].

The process of exosome biogenesis (Figure 1) is divided into three steps: (a) the formation of endocytic vesicles from the plasma membrane, (b) the invagination of the endosomal membrane that results in the formation of multivesicular bodies (MVBs) consisting of intraluminal vesicles (ILVs) and (c) the release of exosomes upon fusion of MVBs with the plasma membrane [24]. In the first step, endocytic vesicles are generated from the plasma membrane, which are then targeted to the early endosome, a distinct membrane-bound endocytic organelle that afterwards matures into a late endosome. The inward budding of the outer endosomal membrane leads to the formation and accumulation of ILVs within the late endosomes, which are termed MVBs. These MVBs then fuse either with the lysosome for degradation or with the plasma membrane, thus releasing ILVs into extracellular space (Figure 1), known as exosomes [25,26]. It is worth mentioning that Rab5GTPases are the biomarkers of early endosomes, whereas Rab7 and Rab9GTPases characterize the late endosomes [27,28].

The formation of MVBs is regulated by processes dependent on the endosomal sorting complexes required for transport (ESCRT), as well as by ESCRT-independent mechanisms. The ESCRT is composed of four soluble multi-complexes (ESCRT-0, -I, -II and -III) and their associated proteins (VPS4, VTA1, ALIX). This complex is responsible for the transportation of membrane-bound ubiquitinated proteins, as well as for endosomal vesiculation [29]. Specifically, the tumor susceptibility gene 101 (TSG101), an ESCRT-I protein regulator of MVB biogenesis, interacts with ubiquitinated proteins and participates in the activation of ESCRT-II complex [30,31], which in turn recruits proteins into MVBs and is responsible for the deubiquitination of cargo proteins prior to their packaging into ILVs. In the final step of ILV formation, ESCRT-III and the vacuolar protein sorting-associated protein 4 (VPS4) ATPase are involved in the dissociation and recycling of vesicles [32,33]. Various mechanisms, including direct membrane fusion, endocytosis or receptor-ligand binding, have been proposed to be involved in the internalization of exosomes by recipient cells, representing thus an important mode of intercellular communication.

### 2.2. Physiological Role of Neuron-Derived Exosomes in Cell-Cell Communication

Neuronal cells secrete exosomes bearing various cell-specific proteins, lipids and RNA transcripts that may affect neurotransmission and protein expression upon their uptake by neighboring neural or glial cells [34,35,36]. It has been reported that exosomes released by N2A neuroblastoma cells are preferentially taken up by astrocytes and oligodendrocytes, whereas exosomes secreted from stimulated primary cortical glutamatergic neurons are preferentially internalized by the presynaptic terminals of hippocampal neurons, indicating that synaptic activation favors the intraneuronal communication via exosomes [37,38]. Moreover, rapid depolarization of human dopaminergic neuroblastoma SH-SY5Y cells evokes a redistribution of small non-coding RNA molecules (miRNAs) involved in synaptic plasticity, a large proportion of which are released via exosomes also containing the synaptic protein MAP1b [39]. Such findings suggest that the observed depolarization-associated redistribution of miRNAs in neurons could be attributed, at least partly, to the release of exosomes enriched in MAP1b in an attempt of the cell to regulate synaptic plasticity. It has been also proposed that the retrograde signaling pathway mediated by exosomes containing Synaptotagmin 4 (Syt4), a membrane trafficking protein implicated in learning and memory, regulates the presynaptic release of neurotransmitters, thus facilitating synaptic growth [40]. Interestingly, active glutamatergic neurons secrete AMPA receptor-associated exosomes, which is considered a homeostatic mechanism for AMPA receptor recycling aiming at the regulation of neuronal firing [34,41]. Finally, subunits of the glutamate receptors, as well as, of other ion channels can be transferred from neuron-to-neuron via exosomes, highlighting further the role of neuron-derived exosomes in the modulation of synaptic plasticity [19].

Apart from the role of neuronal exosomes as mediators of the intraneuronal communication, they can also send signals to glial cells (particularly to astrocytes and microglia) (Figure 1). The exosomal release of various small RNAs and miRNAs from neurons and their internalization by perisynaptic astrocytes has been proven crucial for the regulation of synaptic function and neurotransmission. Such an example is the transfer of the exosome-bound miR-124a from neurons to astrocytes, which increases the protein expression levels of glutamate transporter-1 (GLT1), which prevents neuronal excitotoxicity [42,43]. Other neuron-specific miRNAs that have been identified within exosomes taken-up by astrocytes include miR-669, miR-466, miR-297a-5p and miR-3082-5p, all leading to GLT-1 upregulation [44]. In addition, neuron–microglial communication is also based on the release of neuronally derived exosomes. Strikingly, such exosomes can stimulate microglial phagocytosis of degenerating inactive neurites through the upregulation of complement factors, thus facilitating synaptic pruning [45].

### 2.3. Physiological Role of Glial-Derived Exosomes in Promoting Neuronal Health

The communication between neuronal and glial cells is a crucial determinant for the maintenance of cellular homeostasis in the brain [46], with glia reported to exert both neurotoxic and neuroprotective functions in neurons via the release of extracellular vesicles [47] (Figure 1). A pioneering study by Potolicchio and colleagues in 2005, suggested that exosomes derived from N9 microglial cells contained the aminopeptidase CD13, which is responsible for the degradation of neuropeptides, thus playing a role in neuronal metabolic support and neuropeptide catabolism [20]. Moreover, Tamboli et al. proposed that exosomes secreted from microglia exert therapeutic properties in an Alzheimer’s disease (AD) mouse model, findings which were then further supported by another study reporting that during an inflammatory response in AD, activated and aged microglia release exosomes that aim to restore immune homeostasis [48,49]. Interestingly, microglia-derived exosomes have been shown to transfer various miRNAs to neurons, such as miR-124, which seems to protect neurons against ischemic-brain injury and promote neuronal survival [50]. The findings of another study proposed that extracellular vesicles (EV) secreted from microglial cells stimulated with LPS/IFNγ, contained transcripts of inflammation-related genes, which once injected in glioma-bearing mice, attenuated neuronal cell death and restored brain homeostasis [51]. Furthermore, the exosome-related miR-124-3p has been shown to shift microglia into the anti-inflammatory M2 phenotype and suppress inflammation in injured neurons [52]. It has also been reported that stimulation of Ca^2+^ influx by the ionophore ionomycin or by addition of the excitatory neurotransmitter glutamate in oligodendrocytes, triggers the release of exosomes containing various proteins that are implicated in the trophic support of neuronal axons and in neuronal survival upon oxygen–glucose deprivation [23,53,54]. Furthermore, oligodendroglial exosomes can act in an auto-inhibitory fashion, inhibiting myelin formation [55] or, alternatively, become cleared from the extracellular space by microglia in an immunologically silent manner [56]. Additionally, oligodendroglial-derived exosomes have been proposed to promote axonal transport in starving neurons, whereas mutant exosomes originating from PLP- and CNP-deficient mice seem to lack this ability [57]. 

In regards to astrocytes, it has been demonstrated that upon hypothermia treatment they secrete Hsp70-enriched exosomes, which protect neurons against cellular stress and subsequent death [21]. Other cargo released via exosomes derived from astrocytes includes synapsin I, apolipoprotein D (ApoD) and neuroglobin proteins, all of which have been reported to promote neuronal survival [58,59,60]. Moreover, exosomal miR-200a-3p that suppresses mitogen-activated protein kinase 4 (MKK4), a crucial upstream kinase in the c-Jun N-terminal kinase cell death pathway, displays a neuroprotective role when secreted by healthy, non-stressed astrocytes [14]. In addition, various studies have proposed that miRNAs packed into astrocytic exosomes can regulate the dendritic growth of neurons and exert neuroprotective properties [61,62,63,64,65]. Finally, it is worth mentioning that the cargo of astrocyte-derived extracellular vesicles alters, depending on the stimulus that astrocytes receive i.e., trophic, inflammatory or anti-inflammatory [66,67] thus setting these vesicles as crucial mediators of cell-to-cell communication. 

## 3. A Key Role of Exosomes in Alpha-Synucleinopathies

### 3.1. Neuronal Synucleinopathies

#### 3.1.1. What We Know from PD Cellular Models

Several in vitro studies suggest a potential role of exosomes as mediators of neurodegeneration in PD (Figure 2). Initial studies in αSyn-expressing SH-SY5Y cells proposed that αSyn is partly released via exosomes in a calcium-dependent manner [6] and that such exosomes may mediate αSyn transmission between neuronal cells in a manner dependent upon lysosomal activity [68,69,70]. A subsequent study in primary neurons and neuronal-like cells suggested that exosome-associated αSyn oligomers demonstrate a higher capacity of being taken-up by surrounding cells, as well as, of exerting more toxic effects on recipient cells, compared to exosome-free forms of the protein, uncovering thus a key mechanism in the spread of αSyn pathology in the brain [7]. In accordance with previous observations, it has been reported that exosomes derived from the mouse neural crest-derived Neuro 2A cell line treated with low concentrations of αSyn pre-formed fibrils (PFFs) provide an environment that favors αSyn oligomerization, as assessed by a continuous Thioflavin T fluorescence assay [71]. To further support this notion, the authors analyzed the molecular composition of these exosomes and found that certain exosomal lipids, namely ganglioside lipids GM1 and GM2, are sufficient to cause the accelerated aggregation of αSyn. 

Other studies have shown that not only neuronal- but also glial-derived exosomes may play a significant role in the progression of PD pathogenesis by delivering potential harmful cargoes to neighboring neuronal or glial cells (Figure 2). Various in vitro studies have proposed that microglia, the primary resident immune cells of the brain, are capable of internalizing and subsequently degrading different conformations of αSyn, pinpointing a leading role of microglia on the onset and progression of PD [72,73,74,75,76,77]. On the other hand, pathological αSyn can also activate microglia, leading to the secretion of pro-inflammatory cytokines and neuroinflammation events that may underlie PD pathogenesis [78]. Moreover, microglia themselves can also secrete exosomes [79] and the role of microglial-derived exosomes in cell-to-cell transmission of αSyn pathology has recently gained considerable attention. In particular, treatment of BV-2 microglia cells with aggregated αSyn has been shown to increase the release of exosomes, which may further exert neurotoxic effects due to the expression of the MHC class II molecules and membrane TNF-a [80]. Towards the same direction, inoculation of mouse primary microglia cells with human αSyn PFFs has been reported to evoke the release of exosomes containing pro-inflammatory cytokines that further lead to increased αSyn aggregation and pathology transmission in recipient neuronal cells both in vitro and in vivo [13]. 

Even more, environmental neurotoxic agents such as manganese (Mn^2+^) have been reported to increase neuronal exosome secretion (Figure 2) and promote PD progression [81]. Supportive of this notion are data showing that exposure of MN9D dopaminergic cells to Mn results in increased expression of the Rab27a protein that controls the exosome release. Mn is also suggested to impact extracellular exosome-associated miRNAs that are significantly implicated in pathways found to be dysfunctional in PD [82]. Similarly, exosomes derived from methamphetamine-treated SH-SY5Y cells carry pathological αSyn and exosome-mediated transmission of phosphorylated αSyn at Ser129 from neurons to astrocytes is increased upon methamphetamine exposure [83]. Impairment of the autophagy-lysosome pathway, which is closely linked to PD development, has been reported also to influence exosomal release levels (Figure 2). Different studies have shown that pharmacological inhibition of autophagy promotes exosome release and αSyn transmission [7,84,85]. More specifically, autophagic dysfunction facilitates the release of αSyn via exosomes and the propagation of αSyn-related pathology, which eventually evokes apoptotic cell death of recipient neurons and disease progression [86]. On the contrary, enhancement of autophagy decreases or even inhibits exosome release in cellular models [87]. 

Exosomes, as mentioned above, are also able to shuttle miRNAs and contribute to PD progression by modulating the expression of target genes in recipient cells [88,89]. For example, treatment of primary rat cortical astrocytes with the pro-inflammatory cytokines IL-1β and TNF-a induced the secretion of exosome-associated miR-125a-5p and miR-16-5p that down-regulate the expression of the neurotrophin receptor NTRK3 (TRKC), and its downstream effector protein, Bcl2. Down-regulation of these neuronal targets reduces dendritic growth and complexity, spike rates and burst activity [63]. Moreover, application of secreted astrocytic exosomes on dopaminergic cells increased the protein levels of microvesicle-related miR34a that enhances the susceptibility of SH-SY5Y dopaminergic cells to the PD-linked neurotoxins MPP+ and 6-OHDA through repression of Bcl2 protein [90]. Another study reported that miR-137, which regulates oxidation resistance 1 gene, is upregulated and contributes to the induction of oxidative stress in primary neurons derived from PD mouse models [91]. Moreover, astrocyte-derived exosomal miR-200a-3p has been shown to exert a neuroprotective role through down-regulation of the mitogen-activated protein kinase 4 (MKK4) in SH-SY5Y cells and primary dopaminergic neuron cultures [14]. 

Finally, PD-linked genes have also been associated with the regulation of exosome biogenesis and release. Autosomal dominant mutations in *LRRK2* (Leucine-rich repeat kinase 2) gene are associated with both familial and sporadic PD [92,93]. In particular, LRRK2 is a kinase strongly implicated in the endolysosomal pathway, thus regulating exosome biogenesis via its interaction with Rab5b and Rab7 [94,95]. Notably, alterations of LRRK2 expression in primary neurons significantly impaired synaptic vesicle endocytosis, a defect that could be rescued by the simultaneous overexpression of Rab5b [94]. Moreover, phosphorylated LRRK2 at Ser-1292 has been found associated with exosomes isolated from urine of PD patients, suggesting that the LRRK2-dependent formation of exosomes may be linked to sporadic PD [96,97]. Another gene related to the development of PD is ATP13A2 (PARK9), which encodes for a transmembrane endolysosomal ATPase, expressed in the pyramidal neurons within the cerebral cortex and the dopaminergic neurons of the substantia nigra [98]. ATP13A2 was detected in MVBs, and its presence seems to be related to the number of ILVs and the release of exosomes in general and of exosome-associated αSyn [10,99,100]. Similar effects have been reported for the vacuolar protein sorting 35 (VPS35), a component of the retromer complex involved in the recycling of membrane proteins from endosomes to the trans-Golgi network. The D620N mutation in VPS35 gene has been linked to late-onset PD and has been found to cause endosomal alterations and trafficking defects [101]. In addition, mutations in the E3 ligase parkin, which is physiologically implicated in the regulation of mitophagy through the ubiquitination of mitochondrial membrane proteins, alter the organization of MVBs increasing thus the exosome release [102]. 

#### 3.1.2. Knowledge Acquired from PD Animal Models

Aging is another factor that may play an important role in exosome-associated αSyn transmission, given that microglia derived from elderly mice have shown to exert deficits in the uptake of exosome-associate αSyn, as compared to microglia derived from young animals [103]. Given that aging is the primary risk factor for developing PD, such data insinuate that a larger number of pathology-related exosomes are “trapped” in the intercellular space of aged mice and this impaired clearance of pathological αSyn species may potentially exert harmful effects on neurons and eventually, may contribute to neuroinflammation and disease progression. Supporting further this hypothesis, a subsequent study showed that the uptake of pathogenic αSyn oligomers bound on exosomes triggered the pro-inflammatory activation of microglial cells and led to the secretion of various cytokines and inhibition of the autophagic machinery, eventually resulting in αSyn accumulation [12].

Interestingly, misfolded and oligomeric αSyn, or even the PD-linked A53T αSyn mutant seem to be rather associated with exosomal vesicles, leading to the hypothesis that pathological αSyn species can be spread via their interaction with membrane vesicles [85,104,105]. Moreover, it has been shown that delivery of extracellular vesicles isolated from αSyn-overexpressing HEK293 cells into the striatum of wild-type mice is responsible for the spread of human αSyn throughout the brain [106]. On the contrary, others supported that when exosomes pre-incubated with αSyn pre-formed fibrils (PFFs) were delivered to the striatum of wild-type mice, they neutralized the toxic effects of PFFs, thus raising questions regarding the precise role of exosomes in the development and spread of neuropathological processes [107]. Another study suggested that the release of exosome-associated αSyn is regulated by sumoylation [108]. Recent evidence indicates that exosomes are partially responsible for the transmission of αSyn-related pathology from neurons to astrocytes and that the internalization of αSyn by astroglia induces inflammatory responses [83]. 

Strikingly, treatment of Prnp-αSyn A53T transgenic mice with conduritol-B epoxide (CBE), a pharmacological inhibitor of the lysosomal enzyme β-glucocerebrosidase (GCase), resulted in elevated levels of exosome-associated oligomeric αSyn species [109]. Additionally, GCase deregulation resulted in high levels of extracellular vesicles in the hemolymph of a Drosophila model of PD [110].

#### 3.1.3. Lessons Obtained from Patient-Derived Material

The delivery of exosomes isolated from the brain, CSF or blood (plasma or serum) of patients with alpha-synucleinopathies in various cellular and animal models has shed light on their potential role in the aggregation and seeding of aSyn pathology. Analysis of CSF-derived exosomes from PD patients and controls revealed that only PD and not control exosomes evoke αSyn oligomerization when applied to human H4 neuroglioma cells, probably due to the presence of aberrant protein species that are distinct from those contained in exosomes derived from control subjects [111]. Notably, microglial-derived exosomes have been identified in the CSF of PD and MSA patients, triggering αSyn aggregation in vitro [13]. Additionally, it has been shown that plasma exosomes derived from PD patients can be taken up and subsequent activate microglia in vitro. Patient-derived plasma exosomes were found to inhibit autophagy in BV2 cells and subsequently accelerate aggregation and secretion of αSyn [12]. 

Interestingly, it has been shown that microglial cells were more prone to internalize exosomes in vivo, rather than neuronal or astroglial cells, in an experimental set-up where exosomes derived from PD patients’ plasma were injected in the striatum of 8-month-old mice [12]. Furthermore, it has been reported that neuron-derived exosomes isolated from the brain, CSF or blood of PD or DLB patients can trigger αSyn aggregation in both in vitro and in vivo models [69,111,112,113]. Even more, likely neurogenic L1CAM-purified extracellular vesicles (EVs) derived from the plasma of PD patients have been recently shown to increase the accumulation of αSyn in the midbrain and to accelerate the progression of PD pathology in the Prnp-αSyn A53T transgenic mice [114]. EV-associated miRNA profiling revealed that the novel miRNA, novel_miR_44438, was expressed at significantly higher levels in neurogenic EVs from PD subjects than in healthy controls, inhibiting neuronal exosome release and αSyn efflux through the NDST1-HS pathway [114].

Since exosomes are considered as potential carriers of toxic proteins, their role in PD pathogenesis has been extensively studied via the measurement of their levels in patient-derived biological fluids. Specifically, it has been demonstrated that although the levels of αSyn in the CSF of PD patients were lower compared to controls, the exosome-associated αSyn in the plasma of these patients was significantly higher [115,116,117,118]. As mentioned above, PD pathogenesis is closely linked to lysosomal function, with up to 7% of patients carrying a loss-of-function mutation in the GBA1 gene that encodes for the lysosomal enzyme β-glucocerebrosidase (GCase) [119,120,121]. Finally, it is worth-mentioning that the ratio of exosome-bound αSyn/total αSyn in the plasma of patients with PD is related to GCase enzymatic activity [122]. 

#### 3.1.4. Extracellular Vesicles in DLB

In contrast to the plethora of studies investigating the role of extracellular vesicles (or exosomes in particular) in both the pathogenesis and the prognosis of PD, there is a paucity of data regarding the implication of these nanovesicles in DLB. Only a few studies have focused on the involvement of extracellular vesicles in the pathogenesis of the disease or their role as potential biomarkers for DLB. Specifically, it has been originally reported that CSF-derived exosomes from DLB patients are enriched in αSyn, which can be spread and trigger αSyn oligomerization in a dose-dependent manner [111]. These findings were further supported when one year later it was reported that brain exosomes isolated from DLB patients containing Aβ, tau and αSyn, led to the recruitment of tau and αSyn into the formation of phosphorylated protein aggregates in non-diseased rodent brains [112]. Such exosomes seem to be internalized preferably by neurons and secondarily by astrocytes via a Rab5-mediated endocytosis, thus participating in the establishment of αSyn pathology [112]. Similarly, both PD- and DLB-derived exosomes were found to induce the oligomerization of soluble αSyn in recipient cells [7]. Interestingly, the number of CSF-derived exosomes and the exosome-associated αSyn cargo isolated from DLB patients were found significantly lower when compared to PD patients [111].

DLB and AD are the most common types of dementia and the identification of valid biomarkers that could distinguish between these two diseases is an unmet need. To this end, it has been proposed that the plasma microRNA expression profile could be utilized for the differential diagnosis of the two diseases, given that hsa-miR-451a and hsa-miR-21-5p were significantly down regulated in AD as compared to DLB cases [123]. Likewise, a more recent study demonstrated a significantly reduced expression of various pro-inflammatory genes and the down-regulation of several inflammatory pathways in DLB serum EVs, thus proposing that these vesicles could serve as potential diagnostic markers for DLB [124]. Moreover, in the same study the down-regulation of the protein ubiquitination pathway-related genes UBE3A, USP47, and PSMD4, or even of the ATP-binding cassette family genes ABCA7 and ABCA13 in DLB-derived EVs, further suggested their use as potential diagnostic biomarkers for DLB [124] (Table 1). However, additional studies are required to identify blood-based diagnostic biomarkers for DLB and related synucleinopathies, since various methodological challenges need to be currently addressed.

### 3.2. Exosomes in MSA

MSA, albeit less prevalent, pertains to the broad spectrum of alpha-Synucleinopathies [125]. As distinct from PD and DLB, it is widely acknowledged that in MSA the aggregation of pathological αSyn species takes place primarily in the cytoplasm of oligodendrocytes and secondarily in neurons, thus leading to the formation of glial cytoplasmic inclusions (GCIs) and neuronal cytoplasmic inclusions (NCIs), respectively [126]. Although several efforts have been dedicated to understanding the precise mechanisms prompting αSyn accumulation, propagation and neurodegeneration in MSA, they still remain elusive. However, the release of αSyn by both neuronal and glial cells together with the stepwise spreading of αSyn pathology favors the contribution of the extracellular αSyn as a potential pathogenic ‘prion-like’ agent [127] and the exosomes as a possible mediating pathway of αSyn propagation. Notwithstanding, the significant achievements and existing literature regarding the association of extracellular vesicles, and exosomes in particular, with the progression of PD pathology, it seems that we are only at the tip of the iceberg concerning the implication of these nanovesicles in MSA-related pathology (Figure 2). 

In an attempt to identify blood-based biomarkers to facilitate the differential diagnosis between PD, MSA and progressive supranuclear palsy (PSP) and to monitor disease progression, an initial study assessed the abundance of brain-derived exosomes of neuronal, astroglial or oligodendroglial origin isolated from blood plasma [128]. From this analysis, it was reported that the plasma levels of neuronal-derived exosomes are statistically significantly lower in MSA patients compared to PD patients. In another study, whereby the αSyn cargo within serum neuronal exosomes has been estimated, it has become evident that exosomal αSyn in subjects affected by MSA was unchanged compared to control subjects and concomitantly was two-fold less than that measured in PD patients [129,130]. Both the unaltered amounts of neuron-derived exosomes and the unchanged αSyn levels within these specific nanovesicles in the blood of MSA patients can be associated with the oligodendroglial nature of the disease. Therefore, a subsequent study attempted to untangle the role of oligodendroglial-derived exosomes as a key participant in MSA pathogenesis. More specifically, it was shown that the concentration of oligodendroglial-derived exosomes and αSyn cargo in these exosomes in the plasma of MSA patients were lower as compared to the respective exosomal concentrations in the plasma of age- and sex-matched healthy controls [15]. This difference in αSyn content though was not associated with packaging less of αSyn into MSA-related exosomes since the average αSyn concentration per extracellular vesicle remained unaltered. In accordance with the patient derived data, in vivo data from a PLP-αSyn transgenic mouse model verified the reduced secretion of CNPase-positive oligodendroglial-derived exosomes in the plasma of these mice compared to wild-type mice or the PD-relevant Prnp-αSyn A53T mice. Interestingly, this difference became more apparent as the animals grow up, thus providing a link between age and alterations in exosomal release [15]. Moreover, in the same study it was shown that the release of oligodendroglial-derived exosomes was declined in oligodendrocytes overexpressing human αSyn, or exposed to oligomeric αSyn or treated with GCI-derived αSyn aggregates from MSA patients, as compared to control cells [15]. The mechanism involved was likely related, at least in part, to an αSyn-mediated interference in the interaction between syntaxin 4 and VAMP2, leading to the dysfunction of the SNARE complex and to the impairment of MVB docking into the plasma membrane and the subsequent exosome release [15]. 

In contrast to the previous studies, another report presented conflicting data regarding the exosome-associated αSyn cargo derived from the blood serum of MSA patients [131]. In more detail, they identified elevated concentrations of αSyn within both neuronal- and oligodendroglial-derived exosomes in the blood serum of MSA patients compared to the blood serum of healthy controls and PD patients. Given that MSA patients exhibit higher levels of αSyn in exosomes derived from oligodendrocytes than neurons, whereas the opposite is observed in PD patients, this study combined the average total levels in each cell type into one biomarker and surmised that the ratio of αSyn concentration in oligodendroglial exosomes compared to neuronal exosomes enables to distinguish between the two synucleinopathies, with high sensitivity and specificity [131]. In an attempt to develop a diagnostic model based on plasma-derived subpopulations of EVs in PD, MSA and atypical parkinsonism with tauopathy, a different study reported that immune surface markers in plasma-originated EVs were differentially expressed among PD, MSA patients and healthy controls and thus they could also be utilized as reliable biomarkers for diagnostic purposes [132]. Finally, microglia/macrophage-derived EVs were also proposed to play a key role in MSA pathogenesis and specifically in αSyn spreading. In more detail, it has been shown that such exosomes isolated from the CSF of MSA patients contained elevated levels of αSyn species (both oligomeric and fibrillar) when compared to control samples and it is also noteworthy that they were able to induce αSyn aggregation in recipient cortical neurons [13].

## 4. Exosomes as Disease Biomarkers and Their Potential Role as Nanotherapeutics

Beyond the well-established role of exosomes in the propagation of αSyn-related pathology, neuroinflammation and subsequent neurodegeneration, exosomes may serve as disease biomarkers and targeted exosomes may have the potential for delivering therapeutic agents, including proteins and gene therapy molecules, into the brain. These fascinating facets of exosomes have been mostly described in PD and the data obtained are presented below (Table 1 and Table 2). EVs, including exosomes, have been proposed to act as transporters of potential PD biomarkers (proteins, miRNAs etc.) when isolated from the blood, CSF and urine of patients [133]. Notably, the number of exosomes deriving from the saliva of PD patients was increased when compared to healthy individuals, containing higher levels of oligomeric αSyn [134,135]. Similarly, the proteins SNAP23 and calbindin were enriched in EVs isolated from urinary EVs of PD patients [136]. Interestingly, the levels of the exosome-associated cellular prion protein (PrPC) were elevated in the PD patient plasma samples and were correlated with the progression of the cognitive decline [137]. Furthermore, aggregated αSyn bound on EVs, may serve as a candidate PD biomarker, depending on the origin of these EVs from the plasma, serum or CSF of the patients, since plasma cells are also able to produce such vesicles [138]. 

Moreover, EVs isolated from the plasma of PD patients were shown to carry increased levels of total, monomeric, oligomeric and pSer129-αSyn, with no differences observed in fibrillar αSyn species [12,113]. However, the administration of EVs isolated from the plasma of PD patients to cellular and animal PD models, exhibited both neuroprotective and neurotoxic effects, and new questions arose regarding the role of central versus peripheral EVs [91,139]. Notably, in 2018, Ohmichi and colleagues developed a novel ELISA to distinguish between neuron-, oligodendrocyte and astrocyte-derived exosomes present in the plasma of PD patients, based on the expression of cell-specific protein markers, which may serve as markers of PD staging [128]. Strikingly, it has been recently shown that there is a differential secretion rate of astrocytic EVs that depends on the specific brain area of origin. These EVs seem to play important roles in the restoration of dopaminergic neuronal cell death, via modulating their response to oxidative stress and inflammation [140]. Neuronal survival has also been demonstrated to be promoted by astrocyte-derived EVs carrying apolipoprotein D (ApoD) in diseased neurons, thus further supporting the contribution of such vesicles in intracellular communication and their protective role under pathological conditions [60].

Regarding the CSF-derived exosomes from PD patients, it has been demonstrated that 16 exosomal miRNAs were upregulated, whereas 11 miRNAs were downregulated compared to exosomes obtained from healthy individuals [141]. Furthermore, the levels of miRNA-19b have been found reduced, but miRNA-195 and miRNA-24 were elevated in the serum and CSF of PD patients, thus suggesting the notion that exosomal miRNA profiling may represent an effective method for disease diagnosis [142]. The discovery of both neuronal and glial-derived proteins on CSF-EVs isolated from patients with neurological disorders, revealed the role of neuronal and glial cells both in brain physiology and pathology [143]. In several neurodegenerative diseases, including PD, a dysregulation in the levels of miRNAs has been reported and exosomes may serve as carriers of both miRNA mimics and anti-miRNAs, to restore their physiological levels [141]. The levels of miRNA-155 have been found elevated in both mouse models of αSyn overexpression and miRNA-155 knockout (KO). The absence of miRNA-155 led to MHC class II downregulation and thus reduced inflammatory response and loss of dopaminergic neurons [144]. Furthermore, the delivery of miRNA-7 into the mouse striatum suppressed the activation of Nod-like receptor protein 3 (NLRP3)-mediated inflammasome and decreased dopaminergic neuronal cell death in the 1-methyl-4-phenyl-1,2,3,6-tetrahydropyridine (MPTP) mouse model of PD [145]. Additionally, the levels of various miRNAs loaded in blood-derived EVs from PD patients have been proposed to be indicative of PD-related pathology. Specifically, an enhancement in the cargo of miR-195, miR-24, miR-153, miR-409-3p, miR-10a-5p and let-7-c-3p, miR-331-5p or a decrease of miR-19b, miR-1 and miR-505 in serum or CSF of PD patient-derived exosomes have been reported [141,142,146]. 

Exosomes have been also considered potential carriers of drug targeting against CNS pathologies (Table 2), since they can easily cross the blood brain barrier (BBB) [147]. For example, dopamine-loaded exosomes have been administrated intravenously in a mouse model of PD and surprisingly, were successfully delivered to their brain, attenuating the toxicity induced by free dopamine and displaying therapeutic properties [148]. Additionally, exosomes loaded with curcumin, an anti-inflammatory agent, attenuated the LPS-induced inflammation in mouse brains via their uptake by both activated and resting microglial cells [149]. Exosomes loaded either with the antioxidant protein catalase or with the mRNA of catalase have been also used for PD therapy, finally eliminating ROS production in cell lines and ameliorating motor function in the living mouse brain [150,151,152]. Interestingly, when exosomes loaded with siRNAs or shRNAs against αSyn were injected in PD mice, the levels of αSyn aggregates were significantly reduced and the PD-related symptoms were improved [153,154]. The delivery of siRNAs via exosomes has been extensively studied by numerous groups targeting various genes for the efficient therapy of neurodegenerative disorders and cancer via gene therapy [68,155,156,157]. Moreover, mesenchymal stem cells (MSC)-derived exosomes have been used as potential treatment against CNS pathologies, such as stroke, traumatic brain injury, spinal cord injury and neurodegenerative diseases, such as AD and PD [158,159,160,161,162,163,164]. Accordingly, exosomes derived from stem cells from the dental pulp of human exfoliated deciduous teeth (SHED) have been recently proposed as a novel therapeutic tool against PD, since their intranasal delivery in a rat PD model resulted in improved motor symptoms and increased levels of tyrosine hydroxylase in the substantia nigra and the striatum of the diseased rats [163,165].

Apart from αSyn, various extracellular vesicle-related proteins have been proposed as potential PD-biomarkers. The levels of DJ-1, a protein-sensor of oxidative stress, were found significantly increased in PD plasma-derived exosomes, whereas various mitochondrial markers were decreased in serum exosomes of patients [166,167,168]. Notably, the Raman spectroscopy protocol and proteomics allowed the analysis of circulating EVs and uncovered different proteins EV-cargo, depending on their origin from healthy controls or PD patients [139,169,170,171]. Remarkably, three exosome-associated proteins have been suggested as candidate biomarkers for PD: clusterin, apolipoprotein A1, and complement C1r subcomponent, the levels of which were found elevated in PD patients compared to healthy controls [169]. Accordingly, other studies have revealed that the levels of the serum-derived exosome-related proteins afamin, apolipoprotein D and J, and pigmented epithelium-derived factor were increased, whereas of complement C1q and protein Immunoglobulin Lambda Variable 1-33 (IGLV1-33) Cluster -33 were decreased in PD patients [171]. Additionally, neuronally-derived isolated exosomes were found enriched in insulin signaling proteins, the levels of which were altered upon treatment of idiopathic PD patients with exenatide for various time points [172].

## 5. Conclusions

Exosomes are considered to play a central role in the propagation of pathology in alpha-Synucleinopathies, acting as “*Trojan horses*” that transfer pathological forms of αSyn aggregates between neurons and glia cells and promoting neuroinflammation through the secretion of pro-inflammatory cytokines by activated glial cells, events that may eventually lead to neuronal demise. Research in the field of exosomes has received extensive attention during the last decade, due to findings highlighting their contribution in neuron-glia communication. Moreover, the availability of peripheral biological fluids such as blood plasma, serum, urine and saliva to isolate neuron- or glia-specific exosomes, suggest that exosomes may act as a potential window into neurodegenerative diseases, enabling the dynamic monitoring of ongoing brain changes related to neurodegeneration. Thus, peripheral exosomes circulating in human plasma or serum may provide the basis for the development of such brain-disease related peripheral biomarkers. Moreover, targeted exosomes represent an attractive approach to deliver therapeutic agents for the treatment of neurodegenerative diseases in general.

However, there are still many unknowns, limitations and challenges that need to be addressed before translating the findings obtained from cellular and animal models into clinical therapies. A major challenge in the field is the development of techniques to accurately isolate cell-specific exosomes from the periphery that will enable the better understanding of the unique properties of exosomes in neural–neural and neural–glial cell communication. In addition, as evident from the data presented in the current review, due to the dual facet of exosomes in the brain, identification of the factors that mediate the beneficial or harmful exosomes secretion under physiological and pathological conditions represents an unmet need. In regard to alpha-Synucleinopathies, we believe that the elucidation of the contribution of neural- and oligodendroglial-derived exosomes in the development and progression of PD and MSA, respectively, may provide novel insights for the development of new diagnostic and therapeutic strategies against these similar but distinct brain disorders.

## Figures and Tables

**Figure 1 biomolecules-12-00957-f001:**
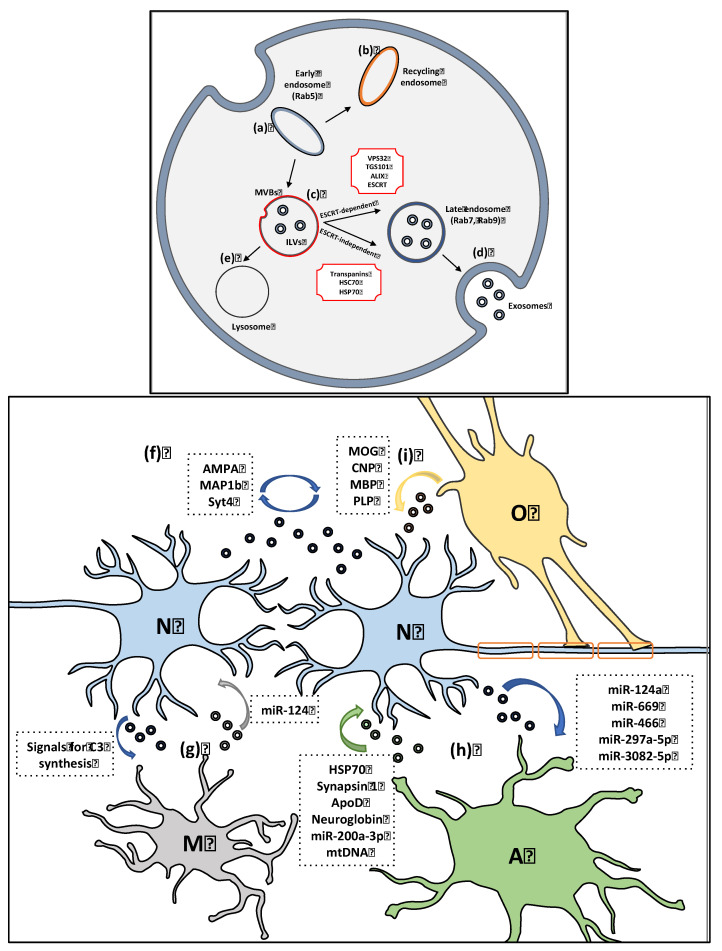
Biogenesis and physiological function of exosomes in neuron-glial communication. (***Up***) (**a**) Endocytic vesicles generate from plasma membrane and are then fused with early endosomes via a Rab5-dependent mechanism. Early endosomes are either sorted for recycling (**b**) or they mature to multivesicular bodies (MVBs) that contain intralumenal vesicles (ILVs) (**c**,**d**) The formation of late endosomes (characterized by the presence of Rab7 and Rab9 GTPases) is regulated by either ESCTR-dependent or ESCRT-independent mechanisms, that finally lead to exosomal release upon the fusion of MVBs with the plasma membrane. (**e**) Alternatively, MVBs fuse with the lysosome, resulting in cargo degradation. (***Bottom***) (**f**) Neuronal-derived exosomes (blue) mediate intraneuronal communication via bearing proteins responsible for neurotransmitter release and synaptic plasticity (AMPA, MAP1b, Syt4). (**g**) Microglia-derived exosomes (gray) transfer various miRNAs (i.e., miR-124) to neurons (N, blue) which promote neuronal survival, whereas neuronal exosomes up-regulate complement factors (C3) in microglia (M, purple), thus modulating synaptic pruning. (**h**) Exosomes secreted from astrocytes (A, green) contain various proteins and RNAs (HSP70, synapsin-1, ApoD, neuroglobin, miR-300a-3p, mtDNA) that protect neurons against cellular stress and death. Exosomes transferred from neurons to astrocytes carry various mRNAs, all of which lead to GLT-1 upregulation, which prevents neuronal excitotoxicity. (**i**) Oligodendrocytes (O, orange) secrete exosomes that are enriched with myelin and stress-protective proteins (MOG, PLP, MBP, CNP) and provide trophic support to neuronal axons.

**Figure 2 biomolecules-12-00957-f002:**
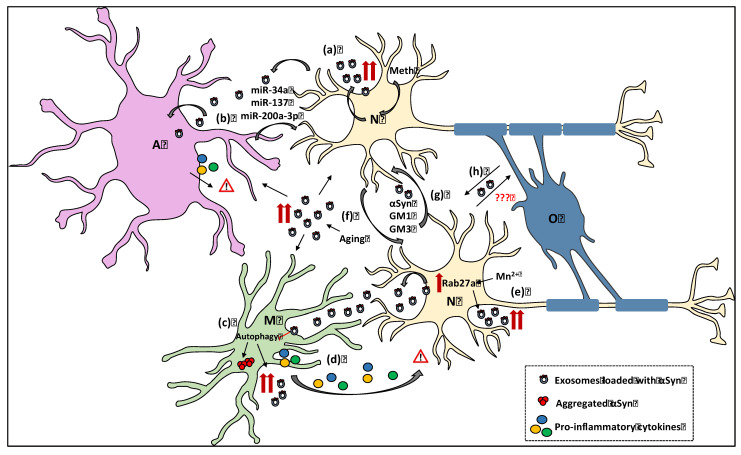
Pathological role of neuronal and glial exosomes in alpha-Synucleinopathies. (**a**) Methamphetamine triggers the release of exosome-associated pathological αSyn from stressed neurons (N, yellow), which is then transferred to astrocytes (A, purple), causing inflammatory responses. (**b**) The astrocytic exosome-related miR-34a and mir-137 induce oxidative stress and enhance the susceptibility of dopaminergic neurons to neurotoxins. On the other hand, miR-200a-3p has been shown to have a neuroprotective role when secreted by astrocytes. (**c**) Internalization of pathological exosome-associated αSyn by microglial cells (M, green) inhibits the autophagy-lysosome machinery, leading to αSyn accumulation within microglia. Moreover, autophagy inhibition enhances further the secretion of exosome-associated αSyn. (**d**) When microglial cells take-up neuronal exosomes they become activated and secrete various cytokines that may have neurotoxic effects. (**e**) Exposure of dopaminergic neurons to metal manganese (Mn^2+^) stimulates the release of exosome-associated αSyn oligomers, due to the enhanced expression of Rab27a. (**f**) Aging negatively affects the uptake of exosomes by microglia resulting in the presence of higher amounts of pathological αSyn-containing exosomes in the intercellular space of older individuals, which may potentially exhibit neurotoxic effects. (**g**) Neuron-derived exosomes transferring αSyn or the ganglioside lipids GM1 and GM2 accelerate the aggregation of intracellular αSyn, thus participating in the spread of neuronal αSyn pathology. (**h**) Although the isolation of oligodendroglia-derived exosomes (O, oligodendrocytes, blue) from blood plasma or serum has been used for the discrimination between PD and MSA patients, the precise role of exosomes in the pathogenesis of MSA (spread of αSyn pathology between oligodendrocytes or between neurons and oligodendrocytes) remains to be determined.

**Table 1 biomolecules-12-00957-t001:** Differential expression of various exosome-associated molecules reported in alpha-Synucleinopathies and their potential role in early disease diagnostics. A table containing the protein, mRNA and miRNA cargoes that are found either increased or decreased within exosomes derived from cerebrospinal fluid (CSF), plasma, serum, urine or saliva of PD-, MSA- or DLB-patients.

	EXOSOMAL BIOMARKERS/EARLY DISEASE DIAGNOSTICS							
	*CSF*		*PLASMA*		*SERUM*		*URINE*	*SALIVA*
	Increased	Decreased	Increased	Decreased	Increased	Decreased	Increased	Increased
**PD**	miR-153	miR-1	PrPc	ATP5A	miR-192	miR-19b	SNAP23	total number of exosomes
	miR-409-3p	miR-19b-3p	monomeric aSyn	NDUFS3	miR-10a-5p	miR-505	CALBINDIN	aSyn oligomers
	miR-10a-5p	*App* mRNA	oligomeric aSyn	SDHB	miR-331-5p	C1q	pSer1292 LRRK2	
	let-7g-3p	*Snca* mRNA	pSer129 aSyn		miR-24	IGLV1-33		
	RP11-462G22.1	*Park7* mRNA	CLUSTERIN		miR-195			
	PCA3	*Fractalkine* mRNA	ApoA1		miR-153			
			C1r		miR-409-3p			
			CD81+SNAP25		let-7g-3p			
			CD81+EAAT1		AFAMIN			
			CD81+OMG		ApoD			
			DJ-1		ApoJ			
					PEDF			
**MSA**	oligomeric aSyn			neuronal-derived exosomes	aSyn in oligo-derived exosomes			
	fibrillar aSyn			oligo-derived exosomes				
**DLB**						*Ubeaα* mRNA		
						*Usp47* mRNA		
						*Psmd4* mRNA		
						*Abca7* mRNA		
						*Abca13* mRNA		

**Table 2 biomolecules-12-00957-t002:** Therapeutic potential of exosomes. A list of the suggested protein and RNA exosomal load of specific cell-derived exosomes used as therapeutic means against alpha-Synucleinopathies (mainly reported in PD). MSC: mesenchymal stem cells; SHED: human exfoliated deciduous teeth.

THERAPEUTIC POTENTIAL	
** EXOSOMAL LOAD **	** CELL-DERIVED EXOSOMES **
**catalase (protein or mRNA)**	**MSC-derived exosomes**
**dopamine**	**SHED-derived exosomes**
**curcumin**	
**Snca siRNAs/shRNAs**	
**mimic-miR-155**	
**miR-7**	

## Data Availability

Not applicable.

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
