# Peer review of "Exosomes in Alpha-Synucleinopathies: Propagators of Pathology or Potential Candidates for Nanotherapeutics?"

_biomolecules, 2022, doi:10.3390/biom12070957_

Round 1

Reviewer 1 Report

Mavroeidi et al. review the role of exosomes in synucleinopathies. In general, the authors do a great job at introducing the relevant biological background of exosomes in brain cell types and reviewing the relevance of exosomes in the different synucleinopathies (Parkinson’s, DLB, MSA). Finally, they explore specifically the biomarkers found in extracellular vesicles.

My only advise it to go through a careful read of the manuscript and make sure that all terms and concepts are properly introduced or explained, even for a readership that may not be familiar with neurobiology.

L88 – expand ILV and MVB

L214 – expand EV

L229 – issue with space, explain what the mouse model is

L359 – expand PFF

L419 – expand GCI

Author Response

We greatly appreciate the reviewer’s comments for our review.

Following his/her suggestion we have reduced the number of abbreviations and introduced or explained the relative concepts properly.

All changes are shown highlighted in the revised manuscript.

Reviewer 2 Report

The manuscripit “Exosomes in alpha-Synucleinopathies: Propagators of pathology or potential candidates for nanotherapeutics?” from Manvroeidi and colleagues represent a brilliant resume of the latest literature about exosomes in synucleinopathies. It summarize both the role of these extracellular vehicles in the neuropathology of this class of disorders but also outline the possible exploitation of exosomes as therapeutic carriers. The paper is well written, all the topics are carefully explained and further explored.

I would like to suggest only minor revisions in order to perfect the manuscript:

-          Abstract: “Beyond neurons, glial cells also release exosomes, which may contain inflammatory molecules and this glia-to-neuron transmission of exosomal alpha-synuclein may contribute to the propagation of pathology and neuroinflammation throughout the brain” – Please add also neuron-to-glia.

-          Figure 1: I would suggest to enlarge the images, especially the right one.

-          Page 6, Line 251-254: “Furthermore, microglial-derived exosomes treated with human αSyn PFFs were reported to induce the release of pro-inflammatory cytokines leading to increased αSyn aggregation and pathology transmission both in vitro and in vivo” – Please rephrase this sentence in order to clarify that the exosomes have been purified from cells that have been treated with PFFs.

-          Paragraph 3.1.1 and 3.12: I would suggest to create a paragraph that is specific for results obtained from patient-derived material.

-          Table 1: Please consider to prepare a separated table for the therapeutic potential of exosomes.

-          Paragraph 4 “Exosomes as disease biomarkers and their potential role as nanotherapeutics” – I would suggest to move the part from line 565 to 581 up (“Apart from aSyn, various EV-related […] depending on their origin from healthy controls of PD patients”), in order to clearly divide the discussion about exosomal cargo as potential biomarker and the use of exosomes as carriers for a therapeutic intervention.

Author Response

We would like to thank the reviewer for his/her valuable comments.

We have followed the reviewer's suggestions and modified all concerns accordingly. All changes are shown highlighted in yellow.